# The Role of TNFR2 and DR3 in the In Vivo Expansion of Tregs in T Cell Depleting Transplantation Regimens

**DOI:** 10.3390/ijms21093347

**Published:** 2020-05-09

**Authors:** Jose-Ignacio Rodriguez-Barbosa, Pascal Schneider, Luis Graca, Leo Bühler, Jose-Antonio Perez-Simon, Maria-Luisa del Rio

**Affiliations:** 1Transplantation Immunobiology, School of Biology and Biotechnology, Institute of Molecular Biology, Genomics and Proteomics, University of Leon, 24071 Leon, Spain; m.delrio@unileon.es; 2Department of Biochemistry, University of Lausanne, 1066 Epalinges, Switzerland; pascal.schneider@unil.ch; 3School of Medicine, Institute of Molecular Medicine, University of Lisbon, Avenida Professor Egas Moniz, 1649-028 Lisbon, Portugal; lgraca@medicina.ulisboa.pt; 4Faculty of Science and Medicine, Section of Medicine, University of Fribourg, 1700 Fribourg, Switzerland; leo.h.buhler@gmail.com; 5Department of Hematology, Institute of Biomedicine (IBIS/CSIC), University Hospital Virgen del Rocio, 41013 Sevilla, Spain; josea.perez.simon.sspa@juntadeandalucia.es

**Keywords:** regulatory T cells, lymphopenia, homeostatic proliferation, transplantation, TNF/TNF receptors, graft-versus-host disease, graft rejection

## Abstract

Regulatory T cells (Tregs) are essential for the maintenance of tolerance to self and non-self through cell-intrinsic and cell-extrinsic mechanisms. Peripheral Tregs survival and clonal expansion largely depend on IL-2 and access to co-stimulatory signals such as CD28. Engagement of tumor necrosis factor receptor (TNFR) superfamily members, in particular TNFR2 and DR3, contribute to promote peripheral Tregs expansion and sustain their survival. This property can be leveraged to enhance tolerance to allogeneic transplants by tipping the balance of Tregs over conventional T cells during the course of immune reconstitution. This is of particular interest in peri-transplant tolerance induction protocols in which T cell depletion is applied to reduce the frequency of alloreactive T cells or in conditioning regimens that allow allogeneic bone marrow transplantation. These conditioning regimens are being implemented to limit long-term side effects of continuous immunosuppression and facilitate the establishment of a state of donor-specific tolerance. Lymphopenia-induced homeostatic proliferation in response to cytoreductive conditioning is a window of opportunity to enhance preferential expansion of Tregs during homeostatic proliferation that can be potentiated by agonist stimulation of TNFR.

## 1. Introduction

### 1.1. The Role of Regulatory T Cells in Transplantation

The initial description of T cells exhibiting suppressive function is attributed to Nishizuka and coworkers. Neonatal thymectomy at day 3 of age, but not at day 7 or later, induced autoimmune disease [1]. In the late 1980s, it was discovered that non-depleting monoclonal antibodies targeting co-receptor molecules, namely CD4, could lead to long-term acceptance of allografts in mice [2,3]. That state of immune tolerance was dependent on CD4^+^ T cells, and it could be induced in new cohorts of T cells, a process known as infectious tolerance [4,5]. Those CD4^+^ T cells stimulated with alloantigens acquired a non-responsive state that could transfer tolerance to the same alloantigen present in third-party recipients [6]. In 1995, Sakaguchi and coworkers identified a subpopulation of CD4^+^ T cells that exhibited constitutive expression of the IL-2R alpha subunit (CD25) and that were termed regulatory T cells (Tregs) [7,8]. The adoptive transfer of CD4^+^CD25^+^ T cells prevented autoimmune disease development induced by CD4^+^CD25^−^ T cells [7], and the suppressive function was defined later as dependent on the expression of transcription factor forkhead box P3 (Foxp3) [9]. Later, it was shown that the antibody-induced transplantation tolerance was also mediated by CD4^+^CD25^+^ cells [10]. Those regulatory cells were demonstrated to seed tolerated allografts, protecting them from rejection [11]. Golshayan et al. showed that in vitro expanded alloantigen-specific Tregs cell clones provide efficient adjuvant therapy to prolong allograft survival [12].

Among Foxp3^+^ Treg cells, some of them are generated in the thymus (tTreg) while others are induced from uncommitted T cells in the periphery (pTreg) when activated in the presence of TGF-β and IL-2 [13]. pTreg cells are essential in the control of inflammation at the epithelial barriers (gut and lung) [14] and in the modulation of persistent chronic inflammation in infectious diseases [15]. In these two scenarios, Tregs dampen inflammation to limit tissue damage and the subsequent immune pathology [16]. These regulatory T cells are also essential to counterbalance the pathogenic allogeneic immune responses in transplantation [11,17,18].

In vitro artificially differentiated Tregs (so called, in vitro-induced Tregs or iTregs) have been the subject of intense research in the clinical arena for their promising therapeutic activity in exacerbated inflammatory diseases, such as autoimmunity, prevention of graft-versus-host disease, graft rejection, and chronic persistent infection. This name refers to all Tregs differentiated ex vivo from naïve T cells using different methodologies such as exposure to signal 1 (anti-CD3)/signal 2 (anti-CD28) in the presence of TGF-β and IL-2 to promote their clonal expansion. However, iTregs lack a proper epigenetic programing and, as a result, tend to be unstable and lose Foxp3 expression [19]. Besides the classical populations of Foxp3^+^ Tregs of lymphoid organs, an emerging diversity of tissue-resident Tregs populations distributed throughout the body and detected in the tumor microenvironment has been identified. It appears that they can gain specific transcription factors typical of different pathogenic T cell subsets and adopt similar migration patterns to follow them to the sites of inflammation where they can modulate in situ the course of the immune reaction [20]. A particular Treg population with specific anatomic preferences is T follicular regulatory (Tfr) cells that have the unique ability to migrate to B cell follicles where they regulate germinal center reaction and humoral immune responses [21].

The number of Treg cells and their suppressive functional activity are of paramount importance for the maintenance of normal homeostasis of the immune system. Tregs differentiation and survival depends on soluble and membrane-bound signals delivered by nearby immune cells and stroma cells of lymphoid and non-lymphoid tissues [14,17,22]. An outlook of the cell-intrinsic and cell-extrinsic mechanisms mediated by Tregs is depicted in Figure 1.

### 1.2. T Cell Depletion in Peri-Transplant Regimens and Allogeneic Bone Marrow Transplantation to Promote Long-Term Survival of Allografts

Current immunosuppression effectively prevents and reverses acute rejection episodes; however, the long-term survival of transplanted organs has not improved significantly in the last two decades. This is in part due to the failure of immunosuppressive drugs to prevent chronic rejection and the intrinsic toxic side effects of the long-term administration of immunosuppression that gradually deteriorate graft function of the transplanted organ resulting in end-stage disease [23,24]. The other clinical scenario is allogeneic bone marrow transplantation for the treatment of malignancies or the induction of transplantation tolerance that requires cytoreductive-conditioning regimens for the successful engraftment of donor hematopoietic cells. In order to tackle these hurdles, tolerance induction based on peri-transplant T cell depletion or conditioning of the host with low-dose irradiation and co-stimulatory blockade have been put forward to minimize immunosuppressants, which are otherwise necessary for the maintenance of graft survival in the long-term [25,26]. In transplantation, the use of polyclonal rabbit serum against thymocytes (rATG) or the use of alemtuzumab monoclonal antibody (anti-CD52, CAMPATH-1), besides irradiation, are the main common approaches to reduce the frequency of alloreactive T cells [27,28]. These approaches, however, induce lymphopenia, which stimulates homeostatic proliferation and restoration of the normal lymphocyte counts in the peripheral compartments of secondary lymphoid organs. This proliferative response converts naïve T cells into memory T cells with increased cytotoxic capacity that then become resistant to current immunosuppressive regimens, including CTLA-4.Ig-mediated co-stimulation blockade [27,29,30]. Henceforth, we will refer as T cell repertoire resistant to costimulatory blockade to the set of T cells arisen after T cell depletion in response to lymphopenia that repopulate the peripheral T cell compartment.

American physiologist Walter Cannon was the first scientist that coined the term homeostasis to define the tendency of an organism to restore its original status in the face of unexpected disturbances. Cell numbers are kept constant despite alterations inflicted by the action of different inflammatory threats [31]. The principle of immune homeostatic regulation is competition for limited resources of the remaining cells left within an emptied niche when T cell numbers are reduced after lymphocyte depletion leading to lymphopenia [32]. Under non-inflammatory conditions, homeostasis of naïve and memory T cells is under independent regulation [32]. The separation of cellular niches ensures simultaneously efficient protection against common pathogens, the maintenance of immunological T-cell memory, and a reservoir of T cell repertoire diversity capable of coping with new antigenic threats [33].

The accomplishment of complete immune repopulation after lymphopenia induction (T cell depletion or irradiation) is still an unmet goal in many of the clinical settings such as in autologous or allogeneic hematopoietic stem cell transplantation (HSCT) and cancer chemotherapy. In both humans and mice, lymphopenia creates an immunological niche of emptied space that is readily repopulated. This situation leads to conversion of residual naïve T cells into memory T cells, which, along with residual memory T cells resistant to T cell depletion, would expand in the periphery giving rise to an oligoclonal T cell repertoire with limited T cell receptor (TCR) diversity and prone to autoimmunity [34,35,36]. The repopulation of lymphocytes in lymphopenic environments depends on the relative contribution of two orchestrating reconstitution pathways. The first path is peripheral expansion of naïve, effector/memory residual T cells that remain after T cell depletion that proliferate actively in response to antigenic stimuli (such as commensal bacteria of the gut), as well as to freely available abundant homeostatic cytokines such as IL-7, IL-2, and IL-15. In the presence of an active thymus, this initial burst of memory T cell expansion is followed by the generation of a wave of recent thymic emigrants via thymopoiesis and their export to the peripheral pool [37,38]. T cell repopulation in response to lymphopenia revealed the existence of two types of lymphopenic environments: severe lymphopenia typical of immunodeficient mice (nude mice, SCID mice, Rag1 KO mice, or lethally irradiated mice) and mild lymphopenia that occurs in normal antibody-mediated T cell depleted mice or in sub-lethally irradiated mice [28,29,31,32]. In severe lymphopenia, an intense, fast, spontaneous homeostatic proliferation predominates, which is IL-7-independent and driven by antigens derived from commensal bacteria of the gut. Cell division occurs every 6–8 h and permits the reconstitution of an oligoclonal low TCR affinity T cell repertoire cross-reactive with endogenous antigens (self-antigens and commensal microflora) and prone to induce tissue inflammation and autoimmune disease [28,29]. In slow homeostatic proliferation, the immune system faces less severe lymphopenia, often induced through selective depletion of T lymphocytes via rATG or alemtuzumab, or after a sublethal dose of irradiation, in the presence of an intact thymus. In this scenario, a slow homeostatic proliferation rate predominates in response to this type of mild lymphopenia. It is IL-7-dependent, TCR-independent, and cell division occurs in secondary lymphoid organs every 24–36 h [37]. Contrary to fast proliferation, slow proliferation is more beneficial for the host as it regenerates a more diverse TCR repertoire composed predominantly of naïve T cells [29]. One limitation of these T cell repertories regenerated after lymphopenia inductive regimens is that they become highly cytotoxic against donor cells, and the frequency of memory-like T cells resistant to conventional immunosuppression and co-stimulation blockade is high [39]. Therefore, these clinical interventions in the early post-transplant stage may predispose patients to rejection due to a misbalance of naïve/memory T cells and Treg cells in the newly developed repertoire [27,30]. These approaches can trigger a serious autoimmune-prone T cell repertoire particularly in adults and in the elderly, in which thymic output is physiologically compromised to some extent [40]. Although thymic export of recent thymic emigrants decays gradually from puberty onwards to adulthood, T cell counts in secondary lymphoid organs remain constant. This suggests that thymic-independent pathways of T cell regeneration that rely on homeostatic proliferation are in place in adults to compensate for the progressive deficiency in thymic output, although this comes at the expense of loss of TCR diversity [40]. In aging people, this scenario creates a bias of the T cell repertoire that predominantly favors T cells exposed to continuous stimulation by the antigenic milieu of the host at the time of T cell repopulation and the loss of TCR diversity. Residual naïve T cells remaining after T cell depletion bearing TCRs for Ags that were absent at the time of T cell reconstitution are barely represented [41] (Figure 2).

The initial, spontaneous, fast peripheral T cell expansion due to active proliferation in response to lymphopenia takes advantage of the presence of an abundant amount of freely available cytokines in the internal milieu and plenty of antigen-driven signals. This abnormal proliferation needs to be regulated at early time points to control the undesirable side effects of a newly generated oligoclonal T cell repertoire mainly composed of effector/memory T cells resistant to immunosuppression in the recipient of an allograft [28,41]. In patients with sufficient thymic function, de novo T cell production and export of recent thymic emigrants (naïve T cells and tTregs) restricts the course of peripheral expansion of memory-like T cells and, thus, helps to restore a diverse T cell repertoire not prone to autoimmunity. This regenerated immune system is fully proficient and operative to recognize and fight infectious threats [33].

Either ex vivo or in vivo expanded Tregs (tTreg or pTreg) represent a unique lymphoid population endowed with cell-intrinsic and cell-extrinsic features to counterbalance the active proliferation of memory-like cells in lymphopenic environments that predisposes the host to autoimmunity and to develop greater alloimmune responses. Therefore, the introduction of novel approaches that allow full reconstitution of the peripheral T cell compartment with sufficient TCR diversity and not prone to autoimmunity needs more intensive research to manipulate lymphopenia-induced homeostatic proliferation with therapeutic purposes to either enhance or mitigate immunological responses. This idea places Tregs at the central stage of current research.

### 1.3. In Vivo versus in Vitro Tregs Expansion to Modulate Graft Rejection

The number of Treg cells and their suppressive functional activity are of paramount importance for the maintenance of normal homeostasis of the immune system. The maintenance of Tregs in vivo depends on CD28 engagement by B7 ligands as demonstrated in B7- or CD28-deficient mice, which are practically devoid of Tregs [42]. This explains why the administration of soluble CTLA-4.Ig (a costimulatory blockade molecule that competes with CD28 for binding to B7 molecules) reduces Tregs cell number in secondary lymphoid organs [43,44,45]. Indeed, there is an inverse correlation between the dose of CTLA4.Ig administered and the absolute number of Tregs in both mice and humans [46,47,48]. In clinical transplantation, nevertheless, the effect of a high-affinity variant of CTLA4.Ig (belatacept) on Tregs has been difficult to assess because this biologic is administered in combination with blocking antibodies targeting IL-2Rα and cyclosporine [48,49]. The latter is known to decrease Tregs survival by blocking IL-2 production. Despite this, it is currently widely accepted that use of high doses of CTLA4.Ig is detrimental to Treg survival, whereas low doses of CTLA4.Ig, unable to saturate B7, may favor Tregs expansion to some extent in the long term [46,47,48].

Another robust costimulatory blockade strategy in transplantation consists of donor-specific transfusion administered along with anti-CD40L blockade (clone MR1, eliminates recently activated CD4 T cells transiently expressing CD40L and prevents T cell help to B cells). This approach is very reproducible and efficient to induce long-term survival of vascularized allografts by inducing T follicular regulatory cells (Tfr) and tipping the balance of Tregs over effector T cells [5]. Tregs apparently increase at the expense of depletion of effector CD4 T cells, but not due to expansion, contributing to modulate T cell mechanisms of rejection [50,51]. When this approach is combined with low doses of CTLA4.Ig, the survival outcome improves, and immunoregulatory mechanisms seem to be involved in tolerance induction [46,51,52].

In brief, the co-stimulation blockade by targeting CD28/B7 and CD40/CD40L pathways is still an unmet clinical reality not comparable with that achieved in preclinical settings. This is in part because anti-CD40L antibody treatment in non-human primate models was associated with severe thromboembolism episodes [53]. Future research will provide us with a further understanding on how to harness these pathways for the benefit of Treg expansion to promote immunoregulatory mechanisms vital for the induction of transplant tolerance.

Since the discovery of the co-culture assay of CD4^+^CD25^+^ (suppressor) T cells and CD4^+^CD25^-^ (effector) T cells, in which the suppressive potency of in vitro derived Tregs could be evaluated by its ability to prevent effector T cell proliferation [54], an array of experimental approaches has been implemented so far for the in vitro expansion of Tregs. Although this is not the focus of this review, it is worth mentioning that, in essence, most Tregs expansion protocols rely on in vitro stimulation of sorted Tregs displaying the CD4^+^CD25^+^CD127^-^ (IL-7Rα) phenotype with polyclonal stimuli that consists of anti-CD3/anti-CD28 antibodies attached to beads. This is done in the presence of growth factors such as soluble IL-2 and/or TGF-β, and rapamycin (the latter to prevent parallel expansion of residual conventional T cells contaminating the cell culture of sorted Tregs) [55,56]. Rapamycin also exhibits an add-on effect expanding Tregs in vitro [57]. Other drugs, such as epigenetic modifiers 5-azacytidine (hypomethylating agent) and decitabine (histone deaminase inhibitor), have been applied in HSCT in an attempt to attenuate GvHD through an epigenetic control of Foxp3 expression in both preclinical and clinical settings. These pharmacological interventions induce a significant increase in the number of pTregs associated with decreased GvHD [58,59]. As opposed to in vitro induced Tregs, which are rather unstable and prone to lose Foxp3 expression and suppressive function after extensive culture periods, in vivo expansion of Tregs opens up an appealing new window of opportunities in the field of immunology that deserves more attention in the coming future.

## 2. Molecules of the Tumor Necrosis Factor (TNF)/TNFR Superfamily Involved in Tregs Expansion

Two main groups of molecules have evolved in the immune system for the maintenance of its function: immunoglobulin superfamily and TNF/TNFR superfamily. While TNFRs are primarily expressed on immune cells, their ligands display a broader pattern of expression not only restricted to immune cells with antigen-presenting function, but also are present in non-immune cells. TNFR molecules are type I proteins that are differentially expressed on Tregs when compared to effector/memory T cells, in which expression is usually lower, although this depends on the TNF/TNFR ligand interaction considered. In contrast, TNF superfamily ligands are type II transmembrane proteins characterized by a C-terminal TNF homology domain (THD), which promotes assembly of homotrimeric molecules and binding to TNF receptor superfamily (TNFRSF) receptors. They exist in two forms: membrane-bound or soluble after proteolytic processing (metalloproteases or other proteases such as furin) of the stalk region separating the THD from the transmembrane domain. They behave as soluble cytokines executing their regulatory or inflammatory functions over the course of the immune response [60,61,62,63]. Agonist antibodies against TNFR or agonist ligands used as soluble fusion recombinant proteins can promote proliferative and survival signals on Tregs. So far, the most promising targetable molecules to achieve in vivo expansion of Tregs are members of the TNFR superfamily, TNFR2 and DR3 and their ligands TNF and TL1A, respectively [64,65,66]. This approach offers some advantages over previous strategies based on the use of IL-2/anti-IL-2 immune complexes because a more selective expansion of Tregs can be achieved [64,65,67].

## 3. Redundant Functional Activity of TNFRSF Members in Tregs Differentiation and Survival

Two signals, signal 1 (TCR/peptide MHC) and signal 2 (CD28/B7), are essential to modulate the initial phase of T cell activation that prepares T cells for peripheral expansion and cell differentiation towards effector T cells. As soon as T cells become activated, profound changes occur in the array of molecules expressed. This response is accompanied by upregulation of cell surface proteins of the tumor necrosis factor receptor (TNFR) superfamily. The most relevant members of this family with implications in Tregs expansion are TNFR2 (TNFRSF1B), DR3 (TNFRSF25), glucocorticoid-induced TNFR-related protein (GITR, TNFRSF18), and OX40 (CD134, TNFRSF4), although in this review we will center our attention on the two first receptors for their primary implication in in vivo Tregs expansion. TNFR members independently or collectively work together to promote Tregs precursor differentiation in the thymus and to maintain Tregs homeostasis and function in the periphery [66,68,69].

Some members of the TNFR superfamily have been reported to play redundant functions in the life of Tregs. Loss of function of each of the above-mentioned TNFR members impairs thymic development and conversion of naïve T cells into peripheral Tregs with suppressive activity. Consequently, Tregs cell numbers in primary and secondary lymphoid organs are reduced [69]. Whereas the genetic deficiency (gene KO or tailless dominant negative forms) of one of the TNFR member results in mild reductions of Tregs cell number, when combined with deficiency of other TNFR members, pTregs counts drop drastically in these mice because of loss of thymic Tregs production from Tregs precursors [68]. This suggests that loss of one TNFRSF member on Tregs can still be compensated for by the function of others. However, the loss of multiple receptors or the loss of downstream signaling molecules they share such as RelA (NF-*k*B transcription factor) leads to severe defects in Tregs formation in the thymus that is translated into a paucity of Tregs in the periphery. This finding supports the primordial function of RelA in Tregs development [70]. In contrast, the transcription factor c-Rel has been shown to be essential for the efficient induction of Foxp3 in thymic Tregs precursors [69]. This suggests a division of labor and cooperation in the transcriptional control of different members of the NF-*k*B transcription factors in Tregs cell development. Despite the relevance of RelA in Treg development, this transcription factor has been shown to be dispensable for the differentiation of antigen-specific CD8^+^ effector T cells as well as CD4^+^ type 1 helper (Th1) and follicular T helper (Tfh) cells [69]. In summary, the TNFRSF–NF-*k*B axis is a non-redundant pathway to maintain Tregs pool constant in lymphoid and non-lymphoid tissues in both mice and humans.

It has been proposed that upregulation of the TNFR members in the thymus and the observed greater expression of these receptors on Tregs than on conventional T cells may be related to positive selection of thymic Tregs. The increased expression of some TNFR members in Tregs cell progenitors of the thymus and their engagement by their correspondent ligands would provide Tregs with a competitive survival advantage over non-Tregs to access a limited source of IL-2 growth factor in the developmental niche of the thymus [69]. These receptors may have a costimulatory effect on Tregs cell precursors and promote their conversion into mature FoxP3^+^ Tregs cells in the thymus [68]. All in all, this would lower the threshold of T cell activation of Tregs that exhibit high-affinity T cell receptors (TCR) and recognize self-peptide ligands in the context of major histocompatibility complex (MHC) class II molecules, and would promote their selection [68,71].

## 4. The Role of TNFR1 and TNFR2 in In Vivo Tregs Expansion

Tumor necrosis factor was first identified and isolated tracking its ability to kill tumor cells in vitro and induce local hemorrhagic necrosis of transplantable tumors in mice, and for mediating endotoxin shock when released in large amounts in response to lipopolysaccharide [72]. It is also of primordial importance in the regulation of leukocyte attachment and migration through the endothelial layer to the site of inflammation [73]. The first key evidence of the role of TNF in immune-related inflammatory disorders came from the finding that antibody-mediated blockade of TNF inhibited the synthesis of pro-inflammatory cytokines such as IL-1, IL-6, and GM-CSF in synovial cell cultures. This discovery fueled the concept that TNF was at the top of a pro-inflammatory cytokine cascade and enlightened the path for treatment development [74].

Tumor necrosis factor (TNF) has two receptors, TNFR1 (TNFRSF1A, CD120a) and TNFR2 (TNFRSF1B, CD120b), each one with different pathways of signal transduction [75]. TNF presents two forms, a membrane-bound form (mTNF that activates both TNFR1 and TNFR2) and a soluble form that activates TNFR1 only. mTNF can be proteolytically processed into soluble TNF (sTNF) by the metalloprotease ADAM17. TNFR1 is widely expressed on any type of cell, although undetectable by flow cytometry on Tregs or conventional T cells. Thus, whereas TNFR1 has ubiquitous cellular expression, TNFR2 presents a more restricted pattern of expression encompassing conventional T cells, Tregs, myeloid-derived suppressor cells (MDSCs), endothelial cells, and neurons [76,77]. Most human and mice Tregs express high levels of TNFR2, which is crucial for Tregs proliferation and the maintenance of Tregs suppressive function [65,78]. TNFR2^+^ Tregs are indeed more potent than TNFR2-deficient Tregs [77,79].

TNFR1, and the closely related receptor DR3, both contain an intracellular death domain (DD) which, according to structural studies on DR3, can shift from a resting compact fold to an active elongated fold able to interact with itself and to additionally bind signaling DD-containing proteins such as TNFR1-associated death domain (TRADD) and RIPK1 [80]. Self-interactions of the DD may extend receptor oligomerization initiated by ligand binding and explain why TNFR1, but not TNFR2 that lack a DD, can respond to soluble TNF (Figure 3A). TNFR1 and TNFR2 signaling has recently been reviewed extensively [81]. Briefly, TRADD recruits a TRAF2-containing ubiquitin ligase complex that modifies RIPK1 with K63-linked ubiquitins, allowing recruitment of a second ubiquitin ligase complex, LUBAC that further modifies RIPK1 with linear ubiquitin chains to recruit the adaptor proteins TAB2 and NEMO and their associated kinases TAK1 and IKK2. TAK1 activates IKK2, which is at the apex of the usually pro-inflammatory and pro-survival classical NF-*k*B signaling pathway (Figure 3B, left part). The signaling complex can detach from the receptor and replace it with the DD-containing adaptor Fas-associated death domain (FADD) that, in turn, recruits pro-caspase-8. When NF-*k*B is active, FLICE/Caspase 8 inhibitory protein (FLIP) expression is induced, FLIP keeps pro-caspase-8 in check, and the soluble complex still activates classical NF-*k*B (Figure 3B, central part). If NF-*k*B is deficient, for example by lack of TRAF2-containing ubiquitin ligase complexes, the soluble signaling complex signals cell death, either by activation of caspase-8 and apoptosis or, in the absence of caspase activity, by RIPK1-mediated, RIPK3- and MLKL-dependent necroptosis (Figure 3B, right part). TNFR2 has no intracellular DD and cannot signal cell death directly. When activated by membrane-bound TNF, TNFR2 directly recruits the TRAF2-containing ubiquitin ligase complex, leading to activation of classical NF-*k*B (Figure 3C, left part). This also activates the alternative NF-*k*B signaling pathway, as the ubiquitin ligase complex is not available anymore to target, via TRAF3, the kinase NIK for proteasomal degradation. NIK accumulates and activates IKK1 at the apex of alternative NF-*k*B signaling (Figure 3C). Depletion of TRAF2-containing ubiquitin ligase complex on TNFR2 can also divert TNFR1 signaling from NF-*k*B to cell death (Figure 3B) [81]. In Tregs, the net effect of TNFR2 signaling is to reinforce expression of signature genes (FoxP3, CTLA-4, CD25, and GITR) and the regulatory function [82].

## 5. Evidence of the Anti-Inflammatory Functions of TNF and Its Role in Tregs Function

Apart from the predominant proinflammatory function of TNF, multiple sources support an anti-inflammatory role of this cytokine in chronic infections and in the chronic phase of many inflammatory diseases [83]. One of the first compelling indications of the impact of TNFR2 in Tregs biology was that TNFR2-deficient mice exhibited a reduction in Tregs cell number in both the thymus and secondary lymphoid organs [84]. This phenotype was similar to that seen in mice that are triple-knockout for the three known (human) TNFR2 ligands: TNF, lymphotoxin-α, and lymphotoxin-β [78], of which TNF might be the most relevant due to the fact that only mouse TNF binds to mouse TNFR2 [85]. Mice knock-in for human TNF, which unlike mouse TNF does not bind to mouse TNFR2, have a decreased Treg function, unless they are also knock-in for human TNFR2. Specific ablation of TNFR2 in Tregs did not decrease their numbers but diminished expression of signature genes and impaired suppressive activity in vitro, and in vivo in an experimental autoimmune encephalitis model, directly demonstrating a critical intrinsic role of TNFR2 to maintain the functionality of Treg cells [82]. Further evidence links TNFR2 with Tregs function because co-transfer of TNFR2-deficient Tregs along with naïve CD4 T cells into immunodeficient Rag1 KO recipient mice did not prevent colitis development [78]. Whereas Tregs are capable to suppress in vivo IL-6 production in response to LPS injection, Tregs from TNFR2-deficient mice failed to do so, although they could still suppress T cell proliferation in the gold standard in vitro co-culture suppression assay [65]. This correlates with the fact that both mouse and human peripheral Tregs expressed remarkably higher surface levels of TNFR2 than non-Tregs, irrespective of their resting or activated status, whereas TNFR1 is barely detectable on T cells or Tregs cells [65]. Another hint in the track for the identification of the regulatory role of TNFR2 came from the observation that polymorphisms of TNFR2 were often associated with increased inflammation in chronic autoimmune diseases and in GvHD (the latter being a common adverse effect of allogeneic bone marrow transplantation) [86]. Moreover, patients with chronic autoimmune diseases or chronic GvHD that were refractory to first-line treatment with steroids and to second-line treatment with TNF blockers, etanercept (TNF decoy receptor) or infliximab (chimeric recombinant anti-TNF antibody), sometimes suffered from worsened symptoms [87]. In the same line of evidence, chronic inflammatory illness and late stages of infection are more severe in TNF-deficient mice than in WT mice [83]. Globally, these findings suggest that blockade of TNF in the chronic phase of the inflammatory response could interfere with natural regulatory mechanisms necessary to halt disease progression.

In vitro studies of Tregs expansion in the presence of IL-2 and proinflammatory cytokines (i.e., TNF) have also contributed to gain insight into the role of TNF on Tregs function. Although some in vitro studies of Tregs exposure to TNF claimed that this cytokine interfered with Tregs suppressive function [88,89], other authors support the notion of beneficial effects on Tregs. In this regard, TNF was shown to stimulate in vitro proliferation and expansion of human Tregs that exhibited enhanced suppressive capacity and more stable phenotype [68,90]. Despite these conflicting data, the current consensus is that TNF plays a positive role in Tregs function. These contradictory data might be due, at least in part, to different proportions of aggregates in TNF preparations that would result in different signaling abilities.

The overall effect of TNF on Tregs seems to be mediated by signaling through TNFR2, which is highly expressed on both human and mouse Tregs when compared to conventional T cells [65,76,91].

## 6. Role of TNFR2 in Tregs Expansion in Allogeneic Bone Marrow Transplantation and Cardiac Transplantation

The adoptive transfer of donor-derived Tregs to conditioned recipient mice attenuates the severity of graft-versus-host disease (GvHD) in different settings of allogeneic bone marrow transplantation across distinct MHC histocompatibility barriers [92,93]. Co-administration of in vitro expanded Tregs stimulated via the direct or indirect antigen presentation pathways at the time of bone marrow transplantation induces transplantation tolerance to skin and heart transplants of the same haplotype and protects them against acute and chronic rejection [94].

In the field of transplantation research, the most convincing data on the role of TNFR2 in Tregs expansion comes from three simultaneous studies in 2016 that took advantage of experimental mouse models of allogeneic bone marrow transplantation across distinct MHC barriers. In one of the studies, Chopra et al. developed a nonameric TNF complex with selective agonist activity on TNFR2, termed STAR2. This compound was composed of three covalently linked TNF trimers in which TNF monomers were mutated to selectively abrogate binding to TNFR1. They used this biological molecule to treat lethally irradiated recipients with the purpose to expand residual Tregs before performing allogeneic bone marrow transplantation across an MHC class I barrier (Figure 4A). This treatment attenuated the course of GvHD, and the effect was abolished in TNFR2-deficient recipient mice or in Tregs-depleted recipient mice [95]. Leclerc et al. demonstrated, in a semiallogeneic bone marrow transplant setting, that TNFR2 expression on donor adoptively transferred Tregs was essential to mediate their immunosuppressive effect, and that the source of TNF to stimulate TNFR2 comes from allogeneic T cells co-transferred with the bone marrow transplant. Thus, blockade of TNFR2 with an antagonist antibody or the use of TNFR2-deficient Tregs could not modulate GvHD [96] (Figure 4B). Finally, Pierini et al. used an allogeneic bone marrow transplant mouse model across a full MHC mismatched barrier. The approach consisted of the adoptive transfer of preconditioned donor-type Tregs that were in vitro exposed to irradiated peripheral blood cells from acute GvHD recipients (cytokine primed) or primed with TNF in the presence of IL-2. This preconditioning of donor Tregs prevented GvHD at an unfavorable Tregs/allogeneic T cells ratio (1:10), while unprimed Tregs did not confer protection to GvHD [97] (Figure 4C).

In solid organ transplantation, the number of studies is still scarce. It is worth mentioning one study that implicated TNF in chronic rejection of heart allografts [98]. One of the pathognomonic signs of cardiovascular chronic allograft rejection is the thickening of the intimal layer of the artery, a lesion composed of smooth muscle cells and extracellular matrix, along with immune cells infiltration produced in response to steady-state inflammation. Lack of both TNFR1 and TNFR2 on the endothelium of donor hearts attenuates graft vascular disease [98].

In conclusion, these data point to the fact that TNFR2 is a promising targetable molecule for the expansion of donor or recipient Tregs that display functional suppressive activity to modulate the course of GvHD across distinct histocompatibility barriers and to mitigate the process of chronic rejection of vascularized allografts.

## 7. TL1A (TNFSF15)/DR3 (TNFRSF25) Pathway

The TNFRSF member termed death receptor 3 (DR3, TNFRSF25) is a type I transmembrane protein with homology to TNFR1 [60]. DR3 engagement by its ligand TL1A (TNFSF15) mediates NF-kB, MAP kinase, and caspase signaling that regulates cell activation, proliferation, and differentiation, but also modulates apoptosis in immune cells [99]. Death receptor 3 is readily inducible upon exposure to allogeneic stimuli on both CD4 T cells (Tregs and non-Tregs) and on CD8 T cells. Apart from T cells, NKT cells and innate lymphoid cells also express DR3. It is also present to a lower extent in B cells, monocytes, and macrophages. In contrast, TL1A, the ligand of DR3 receptor, presents a pattern of expression restricted to antigen-presenting cells (dendritic cells) and endothelial cells [100,101].

DR3 (TNFRSF25) was initially claimed to be preferentially expressed on Tregs and only barely on resting T cells [64]. This notion generated certain controversy in the field that stemmed from the existence of two alternative DR3 mRNA splicing variants and different anti-DR3 hybridoma clones. DR3 variant 1, with an extracellular region exhibiting four cysteine-rich domains, a transmembrane region, and an intracellular region and DR3 variant 3, which lacks cysteine-rich domain 4 in the extracellular region [64,102]. This controversy unleashed a debate that was resolved with the introduction of a novel monoclonal anti-DR3 antibody. This new reagent identified a common epitope on DR3 of Tregs and conventional T cells, concluding that both cell types expressed a similar amount of DR3 on their cell surface at their resting state. The amount of DR3 increased in both cell types upon activation [102].

The expansion of Tregs upon DR3 engagement was first described in a mouse model of allergy. In this model, the administration of anti-DR3 agonist antibody (clone 4C12) or soluble TL1A.Ig recombinant fusion protein (soluble DR3 agonist) stimulated a transient expansion of Tregs. This agonist activity was modestly increased by the addition of IL-2 immunocomplexes (IL-2/anti-IL-2) [64].

Taking advantage of the knowledge gathered from mouse models of allergy, in vivo expansion of donor Tregs with agonist anti-DR3 antibody was achieved before bone marrow cells were harvested for the transplant. This intervention permits the enrichment of donor Tregs in the bone marrow inoculum that would contribute to mitigate the pathogenic impact of donor anti-host alloreactivity [103]. The expansion of Tregs can also be achieved in pre-conditioned recipient mice, but the treatment with anti-DR3 antibody has to be interrupted soon after allogeneic bone marrow transplantation. Once donor anti-host response is initiated and donor T cells become activated, anti-DR3 treatment may trigger undesirable side effects due to co-stimulation of donor alloreactive T cells. This would exacerbate the course of the GvHD and overcome the initial beneficial anti-inflammatory effects of Tregs [104,105]. In any case, this strategy is still awaiting translational application in humans because, currently, anti-human DR3 agonist antibodies are not yet available for the clinic.

In the setting of heart allogeneic transplantation, treatment with agonist anti-DR3 antibody (clone 4C12) resulted in expansion of Tregs Foxp3^+^ to 30–35% of gated CD4^+^ cells. This increase in Tregs was associated with a significant graft prolongation from day 8 to day 17 that correlated with accumulation of Foxp3^+^ Tregs within the graft and a significant reduction of infiltrating inflammatory cells [106].

In summary, DR3 is a targetable receptor that permits the preferential expansion of donor Tregs or residual host Tregs that remain after host conditioning. Anti-DR3 treatment should be implemented before donor anti-host alloreactive T cells become activated.

## 8. Concluding Remarks on the Therapeutic Implication of Targeting TNFR in Inflammatory Diseases

Etanercept (TNF decoy receptor) and infliximab (chimeric anti-TNF antibody) are two examples of the most common TNF blockers approved for clinical use as a second line for the treatment of steroid-refractory GvHD and autoimmune diseases either alone or in combination with methotrexate; however, some patients do not respond, and disease can be exacerbated upon TNF blockade. This undesirable outcome may be the result of inhibition of Tregs suppressive function that prevents membrane-bound TNF binding to TNFR2.

To overcome these hurdles, there is an emerging interest in the development of recombinant biological compounds that can target specifically TNFR2 without engaging TNFR1, to expand in vivo Tregs and potentiate their beneficial modulatory effect on the course of inflammation [95,107]. Considering that selective neutralization of TNF/TNFR1 signaling is beneficial in several inflammatory diseases, if this could be combined with TNFR2 agonist compounds, a reinforcement of current treatments may be accomplished to attenuate inflammatory diseases and graft rejection. Mutated versions of oligomerized TNF compounds such as STAR-2 (multimeric mutated TNF soluble fusion protein with binding specificity for only TNFR2) or fully/partially agonist non-depleting antibodies targeting TNFR2 anchored to cell surfaces or through FcγR are capable to engage TNFR2 and stimulate Tregs function. This stimulatory outcome occurs despite antibodies exhibiting low affinity and regardless of the location of epitope within the molecule [108]. The use of TNFR agonists in patients undergoing T cell-depleted induced-lymphopenia with the purpose of tipping the balance of Tregs over conventional T cells during the course of immune reconstitution in transplant patients would be a pertinent approach to limit long-term administration of immunosuppression and modulate the course of graft rejection or GvHD.

One consideration that should be kept in mind arises from the fact that T cell activation upregulates most TNFR members in both Tregs and conventional T cells, and the engagement of these receptors can contribute to co-stimulation. This co-signal effect would enhance T cell cytotoxic activity or would make effector T cells more resistant to Tregs-mediated suppression [91]. A sense of caution should, however, accompany future clinical trials to prevent co-stimulation of donor anti-host alloreactive T cells, once they are activated, to prevent worsening of the disease. Therefore, the timing for antibody administration in the approved protocol should be finely tuned.

## Figures and Tables

**Figure 1 ijms-21-03347-f001:**
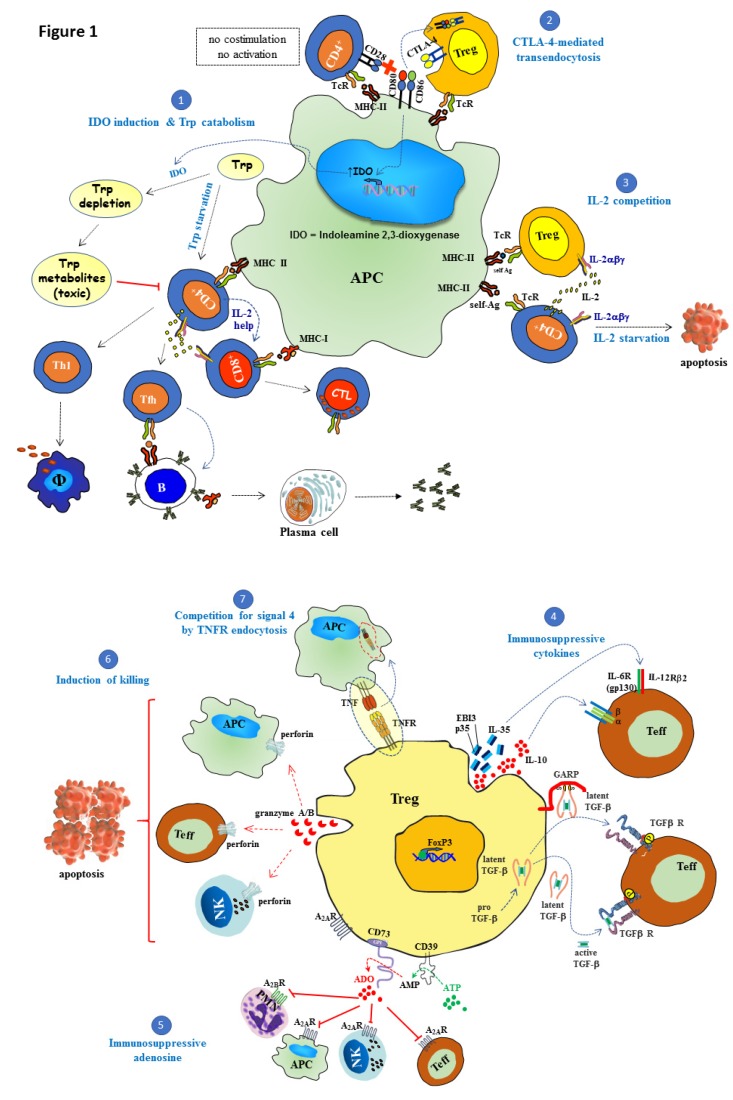
Suppressive mechanisms of regulatory T cells (Tregs). Regulatory T cells (Tregs) play a central role in maintaining immune self-tolerance and preventing autoimmune diseases. Tregs with suppressive function, characterized by expression of the forkhead/winged-helix transcription factor Foxp3 (Foxp3), are generated either in the thymus (tTregs) or in the periphery (pTregs), the latter from naïve T cells under the influence of TGF-β and IL-2. The following cell-extrinsic and cell-intrinsic Tregs suppressive mechanisms operate on effector T cells, NK cells, as well as on antigen-presenting cells (APC) to regulate their function: **(1)** CTLA-4 engagement of CD80/CD86 induces the expression of indoleamine 2,3-dioxygenase (IDO) enzyme. This enzyme degrades tryptophan, which is an essential amino acid required for T cell proliferation and survival. The catabolism of this amino acid gives rise to toxic metabolites that inhibit effector T cell function leading to impaired humoral- and cellular-mediated immune responses. **(2)** An additional cell extrinsic mechanism of suppression is mediated by CTLA-4 that it is constitutively expressed on Tregs. CTLA-4 expressed on Tregs suppresses CD28-mediated T cell activation by a dual mechanism: competing with CD28 for binding to CD80/CD86 (B7.1/B7.2) ligands and induction of CD80/CD86 transendocytosis to shut down co-stimulation. **(3)** Under resting conditions, high-affinity IL-2R (IL-2Rαβγ) is constitutively expressed on Tregs that require basal amounts of IL-2 for their survival. Tregs suppress T cell effector function by limiting the availability of IL-2 produced by activated T cells right after receiving co-stimulatory signal 2, which is essential for autocrine-mediated proliferation. Tregs do not produce IL-2 themselves, although they can compete very efficiently for this cytokine, inducing IL-2 starvation and apoptosis of neighboring effector cells. **(4)** Another suppressor activity of Tregs is mediated via the secretion of immunosuppressive cytokines such as IL-35, IL-10, and TGF-β. IL-35 is a heteromeric member of the IL-12 family composed of p35 (IL-12A) and Ebi3 (Epstein–Barr virus-induced gene 3) subunits, which, together with IL-10, cooperatively suppresses the functional activity of conventional T cells. The other key cytokine in Tregs function is TGF-β which is synthetized as a pro-TGF-β precursor, a heterodimer composed of latency-associated protein (LAP) and mature TGF-β. After signal peptide removal, TGF-β precursor is processed by proteolytic cleavage (furin) to generate the mature polypeptide from the pro-domain. Mature TGF-β remains non-covalently associated to LAP. Latent TGF-β may be secreted alone, but in this form is inactive until it is released from LAP. GARP (glycoprotein A repetitions predominant) is a membrane-bound protein that transports and anchors latent TGF-β to the Tregs cell surface and can inhibit effector function upon cell-to-cell contact with nearby effector T cells expressing the receptor for this cytokine. Active TGF-β binds to type II TGF-β receptor leading to phosphorylation and recruitment of TGF-β receptor type I into a heteromeric receptor complex. Mature TGF-β can act either on Tregs themselves (autocrine manner) or target cells in their vicinity (paracrine manner). **(5)** Adenosine (ADO) is a purine nucleoside produced via enzymatic hydrolysis of extracellular ATP by the coordinated action of nucleoside triphosphate diphosphohydrolase CD39 and the ecto-5′-nucleotidase CD73, two membrane-bound proteins of Tregs. Adenosine interacts with distinct cell surface G-protein-linked receptors expressed on a variety of immune cells. Among these receptors, effector T cells and NK cells express A2A receptors, whereas A2B receptors are found on granulocytes. The engagement of adenosine receptors by adenosine inhibits NK cells, effector T cells functions, and induces a tolerogenic phenotype on APC. **(6)** Tregs can also behave as cytotoxic cells killing target cells (APC, NKs, and T cells) through the secretion of perforin and granzymes, and the subsequent induction of apoptosis. **(7)** Competition for signal 4 by tumor necrosis factor receptor (TNFR) endocytosis: CD27 expressed on the surface of Tregs binds to CD70 on APCs and this leads to CD70/CD27 complex internalization into the APCs, preventing CD70 from binding CD27 present in surrounding cells. Red dotted arrow: Perforin/granzyme cytolytic pathway, blue dotted arrow: cytokine interaction with its receptor, green dotted arrow: enzymatic conversion of ATP to AMP and red dotted arrow: enzymatic conversion of AMP to adenosine.

**Figure 2 ijms-21-03347-f002:**
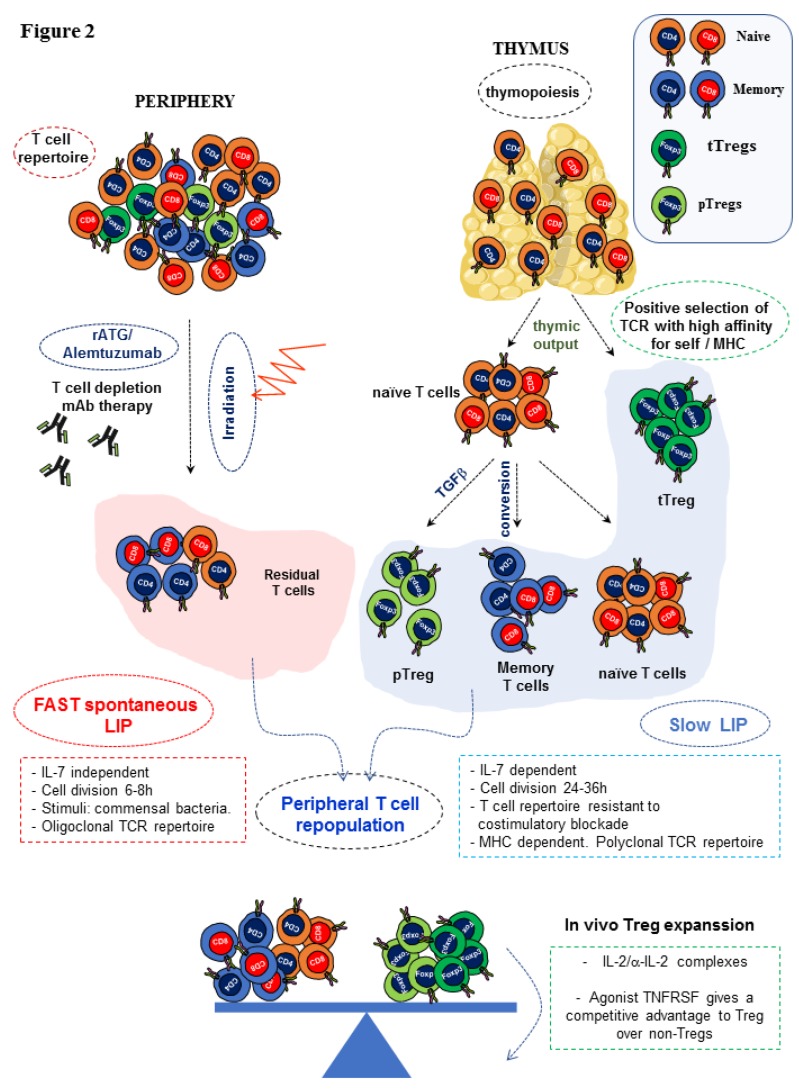
Hypothetical model of lymphopenia-induced homeostatic proliferation (LIP) in response to T cell depletion. Rabbit polyclonal anti-thymocyte globin ATG (rATG) and alemtuzumab are the two biologics most frequently used in T cell depleting protocols to reduce the frequency of alloreactive T cells from the T cell repertoire of patients immediately after transplantation. This clinical intervention aims at preventing early episodes of acute graft rejection and permits post-transplant tapering of immunosuppression. Whole-body irradiation of the recipient is also applied in allogeneic bone marrow transplantation to allow the engraftment of donor stem cells that also depletes the peripheral compartment of T cells. In the depicted hypothetical model, repopulation of the peripheral T cell repertoire occurs via thymus output of recent thymic emigrants and peripheral expansion of predominantly residual effector/memory T cells along with a very small population of naïve T cells resistant to T cell depletion or irradiation. Two waves of T cell repopulation occur in response to lymphopenia to restore a depleted lymphoid peripheral compartment rich in growth factors and space to repopulate. First, the few remaining residual naïve T cells and more abundant residual effector/memory T cells quickly proliferate (fast spontaneous proliferation). After this initial repopulation wave, a second wave of slow T cell proliferation starts. In the presence of an intact thymus, recent thymic emigrants actively exit the thymus and enter the peripheral compartment, in which they are converted into effector/memory T cells while replicating. This wave of thymic output also contains thymic Tregs (tTregs) that, along with pTregs converted from naïve T cells under the influence of TGF-β, configure the Tregs peripheral pool. Thymic emigration is paramount to regulate the rate of recovery of the peripheral T cell pool through a process termed lymphopenia-induced slow proliferation. During this phase of in vivo T cell reconstitution, expansion of Tregs can be favored by the administration of IL-2/anti-IL-2 immune complexes that, combined with agonist signaling through TNFR, provides an advantage to Tregs over alloreactive T cells and tips the balance towards regulation instead of aggressive response. Through this approach of selective expansion of Tregs during T cell repopulation, a more effective modulation of effector alloreactive T cell responses can be achieved in conditioning regimens that require mild T cell depletion in the peri-transplant period. The predominant pathway in the presence of thymic activity is slow proliferation that restores a complete T cell repertoire with sufficient TCR diversity and protects from autoimmunity. Red zigzag arrow: irradiation and blued dotted arrow at the balance indicates the increase in weight of Tregs expansion.

**Figure 3 ijms-21-03347-f003:**
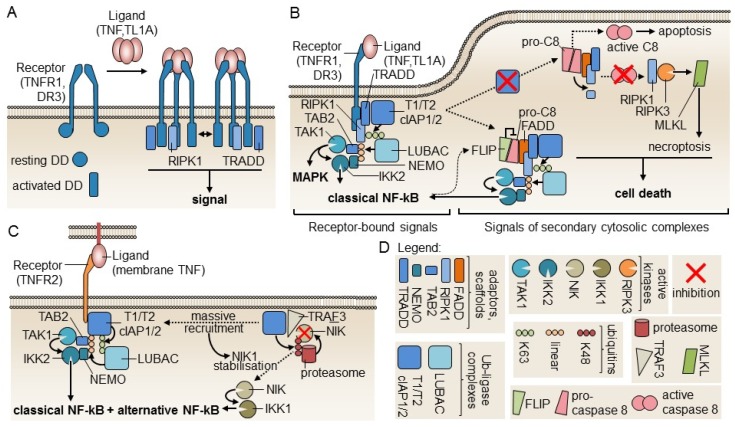
Models of TNFR1 and TNFR2 signaling. (**A**) Upon receptor engagement by ligands, death domains (DDs) in TNFR1 and DR3 may switch from a resting to an activated form that are structurally distinct. Self-interaction of activated DD raises the possibility of larger receptor complexes formation (bi-directional arrow) that would explain responsiveness to soluble ligands. DD-containing proteins TRADD and RIPK1 are also recruited to activated DD. Model based on Yin et al. 2019 [80]. (**B**) TNFR1 receptor-interacting signaling complex and its derived soluble signaling complexes. Successive recruitment of two ubiquitin ligases complexes to TNFR1 allows ubiquitination of RIP1 (K63-linked and linear poly-ubiquitin) to recruit TAK1 and IKK2 for the activation of MAPK and classical NF-*k*B. Unlike what happens in other death receptors, the adapter FADD is not recruited to TNFR1. The signaling complex must first dissociate from receptors as a cytosolic complex to bind FADD and pro-caspase-8. If pro-caspase-8 is inhibited by FLICE/Caspase 8 inhibitory protein (FLIP), cell death is not induced. If NF-*k*B is impaired, active caspase-8 is produced, which processes RIPK1 to an inactive form and induces apoptosis. If caspases are inhibited, RIPK1 is not cleaved and induces necroptosis via RIPK3 and MLKL. Model based on the review of Wajant and Siegmund [81]. (**C**) Signaling by TNFR2. TNFR2 activates classical NF-*k*B by a mechanism similar to that of TNFR1, except that the DD-containing proteins TRADD and RIPK1 are not implicated. As the ubiquitin ligase that targets NIK for degradation in the resting state is recruited to TNFR2 upon signaling, NIK escapes degradation and stimulates alternative NF-*k*B signaling via IKK1. Model based on Wajant and Siegmund [81]. (**D**) Graphical legend for signaling components depicted in panels A, B, and C.

**Figure 4 ijms-21-03347-f004:**
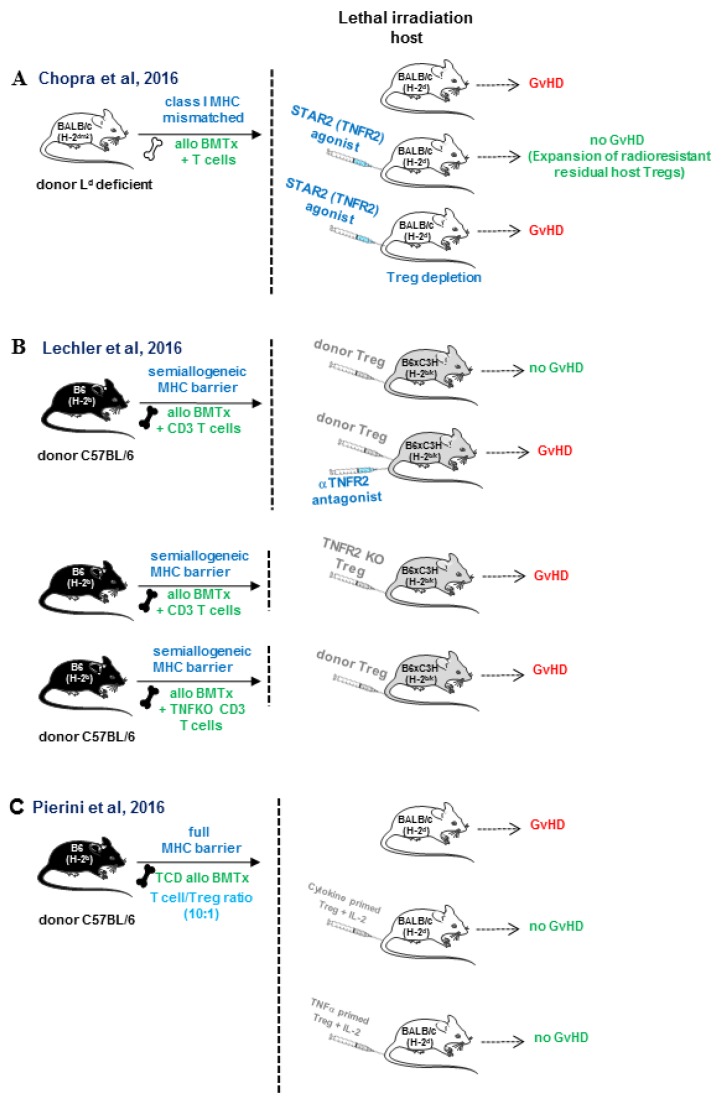
In vivo TNFR2 triggering protects against GvHD by enhancing donor and host Tregs expansion in allogeneic bone marrow transplantation across different histocompatibility barriers. (**A**) Chopra et al. made use of a multimeric recombinant TNFR2 agonist, termed STAR2 containing mutations, to prevent binding to TNFR1. They demonstrated that pretreatment of irradiated recipients with STAR2 peri-transplant expanded residual Tregs that attenuated GvHD and prolonged host survival. This protective effect was abolished in TNFR2-deficient recipient mice and in recipient mice depleted of Tregs [95]. (**B**) Leclerc at al. demonstrated that TNFR2-deficient Tregs or antibody-mediated blockade of TNF/TNFR2 did not abolish the protective effect of Tregs on GvHD. Tregs functional activity depended on TNF production by conventional T cells co-transferred with the allogeneic bone marrow transplant [96]. (**C**) Pierini et al. showed that the preconditioning of donor-type Tregs ex-vivo with irradiated host peripheral blood undergoing acute GvHD (cytokine-primed) or primed with TNF in the presence of IL-2 attenuated GvHD at an unfavorable Tregs: T conventional ratio (1:10) [97]. In contrast, unprimed Tregs did not confer protection to GvHD development in a setting of allogeneic bone marrow transplantation across a fully MHC mismatched barrier.

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
