# Peer review of "The Role of TNFR2 and DR3 in the In Vivo Expansion of Tregs in T Cell Depleting Transplantation Regimens"

_ijms, 2020, doi:10.3390/ijms21093347_

Round 1
Reviewer 1 Report
This is a very well written and interesting review focusing on the role of TNFR family members in regulatory T cell biology. Authors briefly discuss the role of Tregs in transplantation. This is then followed by discussion of the impact of T cell depletion in transplantation in prolongation of allografts survival and subchapters on Treg in vivo and in vitro expansion and role of TNF superfamily in Treg expansion, differentiation and survival. Authors then focus on detailed analysis of role of TNFR1, TNFR2 and TNF itself in Treg function and expansion, followed by the assessment of TL1A/DR3 pathway.
Overall, this is an interesting and well researched manuscript with an original focus and scientific angle. I am sure that journal readers will find the manuscript interesting and useful.
I have a few minor comments:
- Verse 30 – “are being implementing” – please correct to: “are being implemented”
- What do authors mean by “T cell repertoire resistant to costimulatory blockade” as stated in Figure 2 ?
- Could authors provide references for statements in lines 189-202?
- Subchapter In vivo versus in vitro Tregs expansion to modulate graft rejection – authors omitted efforts to expand/generate Tres in vivo by costimulation blockade
- Line 382-383 “TNFR2+ Tregs are indeed the most potent suppressive cells among the population of Tregs” is an overstatement, could you please change to “TNFR2+ Tregs are indeed more potent than TNFR- Tregs”
- Out of curiosity what the authors mean by “non-immune cells of hematopoietic origin” line 297?
- Note, part of Fig 3 legend is obstructed by the Figure itself
- Line 440-443 - consider rewriting this sentence. “This phenotype was similar to that seen in mice that are triple knockout for TNF, lymphotoxin-α, and lymphotoxin-β (the three known human TNFR2 ligands) [67], of which TNF might be the most relevant given that mouse TNFR2 binds …..” - might be easier to follow
- Line 607 “occurs despite antibodies exhibit ” should be “occurs despite antibodies exhibiting ”
Author Response
Reviewer – 1:
Comments and Suggestions for Authors
This is a very well written and interesting review focusing on the role of TNFR family members in regulatory T cell biology. Authors briefly discuss the role of Tregs in transplantation. This is then followed by discussion of the impact of T cell depletion in transplantation in prolongation of allografts survival and subchapters on Treg in vivo and in vitro expansion and role of TNF superfamily in Treg expansion, differentiation and survival. Authors then focus on detailed analysis of role of TNFR1, TNFR2 and TNF itself in Treg function and expansion, followed by the assessment of TL1A/DR3 pathway.
Overall, this is an interesting and well researched manuscript with an original focus and scientific angle. I am sure that journal readers will find the manuscript interesting and useful.
I have a few minor comments:
- Verse 30 – “are being implementing” – please correct to: “are being implemented”
This has been corrected. All corrections are highlighted in yellow
- What do authors mean by “T cell repertoire resistant to costimulatory blockade” as stated in Figure 2 ?
T cell repertoire resistant to costimulatory blockade refers to the set of T cells arisen after T cell depletion in response to lymphopenia that repopulate the peripheral T cell compartment. This clarification was introduced in page 6 and highlighted in yellow.
- Could authors provide references for statements in lines 189-202?
We have added quotes for the statement from line 189-202
- Subchapter In vivo versus in vitro Tregs expansion to modulate graft rejection – authors omitted efforts to expand/generate Tres in vivo by costimulation blockade
I appreciated very much this comment because it was omitted unconsciously. A paragraph highlighted in yellow was introduced into the text to respond to your request. Thanks for your insightful comment.
- Line 382-383 “TNFR2+ Tregs are indeed the most potent suppressive cells among the population of Tregs” is an overstatement, could you please change to “TNFR2+ Tregs are indeed more potent than TNFR- Tregs”
This have been corrected accordingly.
- Out of curiosity what the authors mean by “non-immune cells of hematopoietic origin” line 297?
Thanks for noticing this error. It has been removed from the text.
- Note, part of Fig 3 legend is obstructed by the Figure itself
This obstruction occurred during the layout process of the journal. This has been corrected.
- Line 440-443 - consider rewriting this sentence. “This phenotype was similar to that seen in mice that are triple knockout for TNF, lymphotoxin-α, and lymphotoxin-β (the three known human TNFR2 ligands) [67], of which TNF might be the most relevant given that mouse TNFR2 binds …..” - might be easier to follow
We have rephrased the sentence as follows: This phenotype was similar to that seen in mice that are triple knockout for the three known (human) TNFR2 ligands: TNF, lymphotoxin-α, and lymphotoxin-β [67], of which TNF might be the most relevant due to the fact that only mouse TNF binds to mouse TNFR2
- Line 607 “occurs despite antibodies exhibit ” should be “occurs despite antibodies exhibiting ”
This grammatical mistake has been amended.
Reviewer 2 Report
Authors have written very comprehensive, easy to read and delightfully informative review concerning the role of TNFR2 and DR3 in in vivo expansion of 2 Tregs in T cell depleting transplantation regimens. Paper includes to my knowledge sufficient number of relevant references. Issue has been thoroughly discussed and nice summaries at the bottom of each paragraph makes this complex text very easy to read.
I only have few very minor comments that may or may not be altered in the final paper.
1.
Figure 1. I see no reason to divide the figure into 1.1 and 1.2. One image with two cells and clear bullets showing the site of function discussed in the legend is good enough. Rather long figure legend but serves the paper.
2.
naive > naïve. Many times.
3.
Historic anecdote in lines 162-4 does not have value here (in my opinion)
Author Response
Reviewer-2
Comments and Suggestions for Authors
Authors have written very comprehensive, easy to read and delightfully informative review concerning the role of TNFR2 and DR3 in in vivo expansion of 2 Tregs in T cell depleting transplantation regimens. Paper includes to my knowledge sufficient number of relevant references. Issue has been thoroughly discussed and nice summaries at the bottom of each paragraph makes this complex text very easy to read.
I only have few very minor comments that may or may not be altered in the final paper.
- Figure 1. I see no reason to divide the figure into 1.1 and 1.2. One image with two cells and clear bullets showing the site of function discussed in the legend is good enough. Rather long figure legend but serves the paper.
Your suggestion is very reasonable given that the numbers are consecutive. Figure 1.1 and Figure 1.2 are now under the same epigrapher Figure 1.
- naive > naïve. Many times.
We checked this dieresis issue in a grammar textbook. According to the rule, naïve with dieresis is the correct and official spelling/writing although naive is accepted. We have unified the spelling of this word throughout the manuscript.
- Historic anecdote in lines 162-4 does not have value here (in my opinion)
It is fine. This has been removed from the manuscript.